# The Light Reflection Changes of Monolithic Zirconia and Lithium Disilicate after Using Two External Staining Kits following by Thermocycling

**DOI:** 10.3390/ma16052057

**Published:** 2023-03-02

**Authors:** Eran Dolve, Gil Ben-Izhack, Avi Meirowitz, Hadar Erel, Ofir Rosner, Ameer Biadsee, Diva Lugassy, Asaf Shely

**Affiliations:** Department of Oral Rehabilitation, the Maurice and Gabriela Goldschleger School of Dental Medicine, Sackler Faculty of Medicine, Tel Aviv University, Tel Aviv 6997801, Israel

**Keywords:** thermocycling, CAD-CAM, lithium disilicate, monolithic zirconia, external staining, spectrophotometer, dentistry, prosthodontics

## Abstract

Background: The purpose of the study was to evaluate the changes of light reflection% on two materials (monolithic zirconia and lithium disilicate) after using two external staining kits following by thermocycling. Methods: Specimens were sectioned from monolithic zirconia (n = 60) and lithium disilicate (*n* = 60) then divided into six groups (*n* = 20). Two different types of external staining kits were used and applied to the specimens. The light reflection% was measured before staining, after staining and after thermocycling using a spectrophotometer. Results: The light reflection% of zirconia was significantly higher compared to lithium disilicate at the beginning of the study (*p* = 0.005), after staining with kit 1 (*p* = 0.005) and kit 2 (*p* = 0.005) and after thermocycling (*p* = 0.005). For both materials, the light reflection% was lower after staining with Kit 1 compared to kit 2 (*p* < 0.043). After thermocycling, the light reflection% of lithium disilicate increased (*p* = 0.027) and was unchanged with Zirconia (*p* = 0.527). Conclusions: There is a difference between the materials regarding light reflection% as the monolithic zirconia showed higher light reflection% comparing lithium disilicate throughout the entire experiment. For lithium disilicate, we recommend using kit 1 as we found that, after thermocycling, the light reflection% of kit 2 was increased.

## 1. Introduction

The use of computerized aided design and manufacturing systems (CAD-CAM) to produce chair-side dental restorations has been gaining popularity in recent years [1]. This technology allows dentists and technicians to produce chair-side workflow, including crowns, inlays, onlays and bridges from various materials, such as glass ceramic, zirconia and feldspathic ceramics [2]. Two of the most used materials for single crowns are lithium disilicate and monolithic zirconia. The primary advantage of these ceramic materials is the esthetic outcome, which mimic the natural tooth [3]. 

Both monolithic zirconia and lithium disilicate CAD/CAM blocks and disks are available in different shades and degrees of translucency.

Lithium disilicate blocks, which are widely used, have a unique crystal microstructure, have high strength and high esthetic properties due to its translucency [4]. This material is initially milled in a partially crystallized form and, after the crystallization process, it reaches its ultimate strength and esthetic potential [1,4,5].

In recent years, monolithic zirconia underwent modifications in the manufacturing process to improve its esthetic results. These modifications are conducted by reducing the aluminum content and grain size, an increase in density, and adding the cubic zirconia phase. These modifications improved translucency and lowered light scattering but reduced the material strength [6,7]. 

Today, the clinical success of a prosthodontic treatment relies significantly on the esthetic outcome and its color stability. The definition of color stability in the literature is defined as the ability of the color to reflect the same wave lengths over the years [8].

Color, as we perceive it using the rods and cones in our eyes, resembles the electromagnetic radiation an object reflects in a specific wavelength, while absorbing the radiating visible light. When the wavelength is within the visible spectrum, ranging from 380 nm to 700 nm, the human eye can see it and perceives it as a color [9]. The Munsell system describes color by three parameters: value, hue and chroma. Value describes the overall intensity of how light or dark a color is and is the only parameter that can exist by itself. Chroma is the strength and dominance of the color. Hue, the last parameter, is the color itself [10]. In their study, Todorovie’ et al. [11] found that if the value and chroma are correct, the restoration will be clinically acceptable even if the hue is not exact.

Today, color determination can be performed either instrumentally, using spectrophotometers, digital cameras, spectroradiometers and colorimeters, or conventionally, using the human eye and shade guides [12,13]. 

Visual shade determination is the method most frequently used by dentists despite its high subjectivity, which can lead to inaccuracies [14]. On the other hand, technology-based color matching techniques can be more accurate [11]. These instruments use the CIELAB color system, that is expressed in terms of three coordinate values (L*, a*, b*). The L* represents the brightness of a color, a* represents the red-green axis and b* value represents the yellow or blue chroma [15]. The use of spectrophotometry for measuring light reflection or transmitted light from solid materials such as zirconia or lithium disilicate is well known in the literature [16,17]. 

There are two main staining procedures that are discussed in the literature: internal and external staining methods. In this study, we focus on external staining, which is performed post-crystallization by staining the material with metal oxide (Iron (Fe_2_O_3_), Erbium (Er_2_O_3_), Praseodymium (Pr_6_O_11_) and Cerium (Ce)) solutions. [18,19] Despite the widespread use of external staining kits, the knowledge regarding their long-lasting durability, wear resistance in the oral cavity and the effect they have on light reflection is limited. Unfortunately, no studies have been published regarding the exact components of each kit and the knowledge known to us today is based on the manufacturer guides found on a company’s official website.

The oral cavity is dynamic and has some parameters that influence color stability. These include humidity, the chemical content of foods and fluids in the mouth, stress, forces and temperature [20,21].

Thermocycling resembles thermal stresses induced by water and any solubility effect of neutral water storage on the ceramic tinting glaze. It consists of two baths of water, one at five degrees Celsius and the second fifty-five degrees Celsius. It is estimated that specimens undergo ten thousand cycles, representing one year in the oral cavity. Changes may occur that affect the external surface of the material, and this may lead to a change in the light reflection [21,22].

The coefficient thermal expansion (CTE) of lithium disilicate and monolithic zirconia is well known in the literature (9.5 (10^−6^ K^−1^) and 10.0 (10^−6^ K^−1^), respectively) [23]. The coefficient thermal expansion of the two new staining kits is unknown and we could not identify it in the current literature. As we do not know the difference of the CTE between the materials and the new staining kits, it is very interesting to examine if, after thermocycling, which may affect the external staining, there will be a change in the light reflection%. The purpose of this in vitro study was to evaluate the changes of light reflection% on two materials (monolithic zirconia and lithium disilicate) after using two external staining kits followed by thermocycling. The first null hypothesis was that there will be no change in the light reflection% regardless of the materials and staining kits we used. The second null hypothesis was that there will be no change in the light reflection% before and after thermocycling with both staining kits and materials.

## 2. Materials and Methods

The study was conducted at Tel Aviv University at the School of Dental Medicine with collaboration of the Center of Nanoscience and Nanotechnology. For the purpose of this research, sixty specimens were designed with a 2.4 mm diameter and 0.6 mm thickness using dental computer aided design software (EXOCAD^®^ Dental CAD 2.3 Valentia, Darmstadt, Germany) and milled (M2, Zirconia, Gais, BZ, Italy) from monolithic zirconia disks (Starceram^®^ Z-Smile A2 3Y-TZP-A, H.C. Starck Ceramics Gmbh, Selb, Germany). Another sixty specimens were pressed from lithium disilicate ingots (IPS e.max ceram A2 MO, Ivoclar Vivadent, Schaan, Liechtenstein) into a model with a 2.4 mm diameter and 0.6 mm thickness.

All the specimens were measured using a digital caliper (Mastercraft Electronic Caliper, Canadian Tire Corporation Ltd., Toronto, ON, Canada) to ensure that diameter and thickness were uniform and suitable for the spectrophotometer machine. This allows the light to be reflected on the same area in all specimens (Figure 1).

Each specimen was analyzed using a spectrophotometer machine (Cary 5000-UV-Vis-NIR, Agilent Technologies, Santa Clara, CA, USA) to measure the total reflection%. The specimen holder was designed in a manner that ensured that all the specimens were exposed to the light at the same location throughout the study period. The calibration parameters of the spectrophotometer included a slit of 1.5 nm, a scan speed of 600 nm/min, light range of 300–900 nm (visible and ultra-violet) with a data interval of 1 nm, resolution 0.02/nm and dwell time of 0.1 s. We used an internal diameter of 110 mm. For calculation, we used the wavelengths in the range of 380–700 nm (visible light). The Integrated Sphere accessory can measure both the diffused and specular light waves, meaning that no light will be lost in our measure. In order to calibrate the integrated sphere machine, a baseline measure was taken using a designated white disc (Agilent Technologies, Santa Cruz, CA, USA) and a measure with a designated black disc used to block the integrated sphere. This allowed us to take into consideration how much light is reflected due to the machine itself and not related to the specimens.

For each specimen, 600 hundred results were provided by the machine (for every wavelength at the range of 300–900 nm) and we took a mathematical average from 380–700 nm to measure the light reflection%. As will be described in subsequent paragraphs, the light reflection% of the specimens was measured at the beginning (T0, post crystallization), after staining (T1) and after thermocycling (T2, ten thousand cycles). The spectrophotometer was calibrated before each measurement. It is worth noting that each group was placed separately in the baths and a new group was placed only after the previous group finished ten thousand cycles and the water in the baths was changing. This ensured that each group was submerged in fresh water with no possibility of contamination by reminders of staining materials from previous groups. The two external staining kits used for both lithium disilicate and monolithic zirconia are Stain and Glaze-Crystal Stain Kit (Kit 1) (Ivocolor Vivadent, Schaan, Liechtenstein), which is a paste, and Stain and Glaze Kit (Kit 2) (Dentsply Sirona, Milford, DE, USA), which is a powder and liquid. The staining was conducted according to the manufacturer’s guidance by the same very experienced dental technician (master technician with experience of twenty-five years), using the same paintbrush, ensuring the application of two layers as homogeneously as possible.

The specimens were divided into six groups, with each group consisting of twenty specimens (Table 1).

After the staining process, the discs were put in the dental furnace (Programat^®^ CS, Ivoclar Vivadent, Schaan, Liechtenstein); each material was treated using a compatible program as recommended by the manufacture. They were later polished using the monolithic zirconia polishing kit (RD-HP-KIT1, Strauss and Co, Ra’anana, Israel) and the lithium disilicate polishing kit (RD-RA-EKIT1, Strauss and Co, Ra’anana, Israel). The protocol used for the polishing process recalls that the discs must first be polished with the coarse diamond polishers for removing and shaping, the medium polishers for smoothing the surface and with the fine polishers for a high luster.

After the staining process and light reflection measurements, performed in the spectrophotometer, were completed, thermocycling was applied to all six groups. All specimen groups were subjected to ten thousand thermocycles. The specimens were immersed in each bath for eighteen seconds; the transfer time was eight seconds. The samples were held with a metallic grid (touching only the custom holder) to keep them submerged. The following sequence was used: 5 °C to 55 °C to 5 °C. Then, the specimens were taken to the spectrophotometer for a final calculation of the light reflection displayed after external staining and thermocycling.

To describe the outcome variable of our study (light reflection%), we computed 95% confidence intervals for each of our study groups (2 materials, 3 staining conditions) at the beginning (T0), after staining (T1) and after thermocycling (T2). Box plots were used to illustrate the results.

The first null hypothesis of our study was that there will be no change in the light reflection% regardless the materials and staining kits we used. Our second null hypothesis was that there will be no change in the light reflection% before and after thermocycling with both staining kits and materials. The assumptions of the normality of the variables (light reflection% at the beginning of the study (T0), Light reflection% after staining (T1) and light reflection% after thermocycling (T2) were assessed using the Kolmogorov–Smirnov tests. For comparisons between zirconia and lithium disilicate materials regarding light reflection% at the beginning of the study (T0), after staining with kit 1, kit 2 and after thermocycling, Mann–Whitney tests were performed. For each material, Mann–Whitney tests were performed for comparing light reflection% of samples from samples that did not undergo staining, kit 1 and kit 2 after staining (T1) and after thermocycling (T2). For each material and kit, Wilcoxon Signed Ranks tests were performed to evaluate the differences in light reflection% of each kit at three time periods: T0, T1 and T2.

All analyses were performed using SPSS version 20. Significant statistical differences were defined as *p* < 0.05.

## 3. Results

Light reflection% is reported as percentage values between 0% (totally transparent) and 100% (totally opaque) for visible light, 380–700 nm.

Before analyzing the data, we performed Kolmogorov–Smirnov tests, which showed no normal distribution for the dependent variables in the study (*p* < 0.05): light reflection% at the beginning of the study (T0), Light reflection% after staining (T1) and light reflection% after thermocycling (T2).

The light reflection% median of lithium disilicate and zirconia at the three time periods is showed in Table 2.

### 3.1. Differences between Lithium Disilicate and Zirconia

Significant difference was found in light reflection% at the beginning of the study (T0) between the lithium disilicate (*n* = 60, Mean = 41.20, SD = 5.57) and zirconia (*n* = 60, Mean = 61.18, SD = 3.89), *p* = 0.005. This shows that reflection of visible light by zirconia was higher compared to lithium disilicate.

Significant difference was observed in light reflection% after staining (T1) with kit 1 between the lithium disilicate (*n* = 20, Mean = 26.95, SD = 2.47) and zirconia (*n* = 20, Mean = 38.43, SD = 4.37), *p* = 0.005. The same trend was observed in light reflection% after staining (T1) with kit 2 between the lithium disilicate (*n* = 20, Mean = 28.88, SD = 2.94) and zirconia (*n* = 20, Mean = 52.8, SD = 4.16), *p* = 0.005 (Figure 2). After staining with both kits, the zirconia showed higher reflection of light compared to lithium disilicate.

Significant difference was observed in light reflection% after thermocycling (T2) with kit 1 between the lithium disilicate (*n* = 20, Mean = 27.90, SD = 2.97) and zirconia (*n* = 20, Mean = 38.10, SD = 4.14), *p* = 0.005. The same trend was observed after thermocycling (T2) with kit 2 between the lithium disilicate (*n* = 20, Mean = 34.40, SD = 5.46) and zirconia (*n* = 20, Mean = 51.8, SD = 4.09), *p* = 0.005 (Figure 3). After thermocycling with both kits, the zirconia showed higher reflection of light compared to lithium disilicate. Interestingly, after thermocycling with kit 1, the light reflection% of both materials remains approximately the same. However, after thermocycling with kit 2, the light reflection% of the zirconia remains the same but was increased for the lithium disilicate.

### 3.2. Differences within Lithium Disilicate Group

Significant differences were found in light reflection% between kit 1 and kit 2 after staining (Mean = 26.95, 28.88; SD = 2.47, 2.94, respectively), *p* = 0.043, and after thermocycling (Mean = 27.90, 34.40; SD = 2.97, 5.46, respectively), *p* = 0.0005. In both time periods, kit 1 showed lower light reflection% when compared to kit 2.

Significant differences were found in light reflection% between kit 1 and baseline after staining (Mean = 26.95, 39.93; SD = 2.47, 5.39, respectively), *p* = 0.005, and after thermocycling (Mean = 27.90, 33.78; SD = 2.97, 2.49, respectively), *p* = 0.0005.

Significant differences were found in light reflection% between kit 2 and baseline after staining (Mean = 28.88, 39.93; SD = 2.94, 5.39, respectively), *p* = 0.005. Interestingly, after thermocycling (Mean = 34.40, 33.78; SD = 5.46, 2.49, respectively), no significant difference was found, *p* = 0.640.

Friedman’s Twoway analysis test was performed to evaluate the differences in light reflection% of kit 1 at the three time periods: T0, T1 and T2. Significant differences were found in light reflection% from the beginning (T0), after staining (T1), *p* = 0.0005, and from the beginning (T0) and after thermocycling (T2), *p* = 0.0005. However, no significant differences were found in light reflection% after staining (T1) and after thermocycling (T2), *p* = 0.527.

Friedman’s Twoway analysis test was performed to evaluate the differences in light reflection% of kit 2 at the three time periods: T0, T1 and T2. Significant differences were found in light reflection% from the beginning (T0) and after staining (T1), *p* = 0.0005, from the beginning (T0) and after thermocycling (T2), *p* = 0.002, and from after staining (T1) and after thermocycling (T2), *p* = 0.027.

### 3.3. Differences within Zirconia Group

Significant differences were found in light reflection% between kit 1 and kit 2 after staining (Mean = 38.43, 52.81; SD = 4.37, 4.16, respectively), *p* = 0.0005, and after thermocycling (Mean = 38.10, 51.84; SD = 4.14, 4.09, respectively), *p* = 0.0005. In both time periods, kit 1 showed lower light reflection% when compared to kit 2.

Significant differences were found in light reflection% between kit 1 and baseline after staining (Mean = 38.43, 63.67; SD = 4.37, 2.35, respectively), *p* = 0.0005, and after thermocycling (Mean = 38.10, 62.71; SD = 4.14, 2.44, respectively), *p* = 0.0005.

Significant differences were found in light reflection% between kit 2 and baseline after staining (Mean = 52.81, 63.67; SD = 4.16, 2.35, respectively), *p* = 0.0005, and after thermocycling (Mean = 51.84, 62.71; SD = 4.09, 2.44, respectively), *p* = 0.0005.

Friedman’s Twoway analysis test was performed to evaluate the differences in light reflection% of kit 1 at the three time periods: T0, T1 and T2. Significant differences were found in light reflection% from the beginning (T0) and after staining (T1), *p* = 0.0005, and from the beginning and after thermocycling (T2) *p* = 0.0005. However, no significant differences were found in light reflection% from after staining (T1) and after thermocycling (T2), *p* = 0.527.

Wilcoxon signed ranks tests were performed to evaluate the differences in light reflection% of kit 2 at the three time periods: T0, T1 and T2. Significant differences were found in light reflection% from the beginning (T0) and after staining (T1), *p* = 0.0005, and from the beginning and after thermocycling (T2) *p* = 0.0005. However, and in comparison to lithium disilicate, no significant differences were found in light reflection% from after staining (T1) and after thermocycling (T2), *p* = 0.527.

## 4. Discussion

Our first null hypothesis was rejected, as we found that there was a significant difference in the light reflection% between the two materials at T0 and at T1 for both staining kits. Our second null hypothesis was partially accepted, as there was a significant difference in the light reflection% for the lithium disilicate with kit 2 after thermocycling. For other materials and kits, there was no difference in the light reflection% after thermocycling.

In our study, the external staining stability was correlated to the light reflection% as we compared the light reflection% for each specimen before staining (T0), after staining (T1) and after thermocycling (T2).

A thickness of 0.6 mm was chosen for the experiment since studies show that there is an exponential relationship between the translucency of a material and the thickness in both lithium disilicate and monolithic zirconia. As the thickness decreases the translucency increases. The thickness tested in this study was 0.6 mm for both materials; this did not affect the mechanical and esthetical properties [24].

Various studies have investigated the effects of thermocycling on monolithic zirconia and lithium disilicate. However, publications discussing whether the staining procedures affect the behavior of the restorative materials are lacking [1,19,22].

Previous reports evaluated the effect of acidic drinks on flexural properties and hardness of composite materials, showing a significant effect. However, further studies are needed on the topic in order to test different materials and techniques currently available [25,26].

Hamza et al. examined the effect of mechanical brushing on the color stability and surface roughness of lithium disilicate. In their study, they showed no significant difference in the staining stability after mechanical brushing. In our study, we did not measure surface roughness, that may affect light reflection%. However, we used thermocycling, which represents the oral environment [27]. Kim et al. examined the change in light reflection after several applications of staining and they concluded that both the lightness, yellow chromaticity and opalescence of monolithic zirconia can be changed, but translucency cannot be changed, by the staining procedures [28]. This contrasts with our study, as we showed that the staining procedure reduces the light reflection% for both materials; for the zirconia, the light reflection% reduced significantly when using kit 1 compared to kit 2. After thermocycling, there was no significantly difference between kit 1 and kit 2.

A study by Yuan that examined the effect of brushing and thermocycling on monolithic zirconia and lithium disilicate showed that, for both materials, the color changes were below the selected clinical perceptible threshold [29]. These findings are partially in contrast to our study, as we found that, for lithium disilicate after thermocycling with kit 2, there was increase in light reflection%, showing that the staining was partially removed from the material.

Another issue that must be discussed is that the transmitted light affects translucency and the degree of conversion when using resin cement. A study by Supornpun et al. showed that both the shade and the thickness of zirconia and lithium disilicate affects the translucency, which in turn impact the degree of conversion in resin cement. In our study, the thickness was constant (0.6 mm) but the staining kits led to a decrease of the light reflection%, affecting the translucency of the materials; this may impact the degree of conversion when using resin cements [16].

A study by Farzin has showed that, when using external staining on zirconia, the translucency increased after glazing. This consistent with our study as we also showed that the light reflection% decreased when using a glaze of two different external staining kits [30].

When discussing the value, we know from previous studies that, when using external staining glaze, the value decrease [31,32]. In our study, we showed that, after glazing, the light reflection% decreased, meaning that the value also decreased.

A study by Aljanobi and Al-Sowygh showed that, after 10,000 cycles of thermocycling, the translucency of lithium disilicate and zirconia decreased [33]. In our study we did not see a significant change in the light reflection% after thermocycling (10,000 cycles); the only exception was when using kit 2 on lithium disilicate. We may assume that the glaze was partially removed in this case. According to the results, we may assume that the CTE of the staining kits and the materials is quite similar, except for lithium disilicate and kit 2. It is worth noting that we do not know the CTE of the two staining kits, either from the manufacture or from the literature. When grinding lithium disilicate, we increased the roughness surface; the smoothness of glazed lithium disilicate does not restore by polishing kits [34]. As we polished the lithium disilicate after glaze, we did not try to smooth the surface but to make the glaze more uniform.

Our study is novel in that it assesses the potential change in the light reflection% of two different materials and two different staining kits after thermocycling, thereby mimicking the oral environment.

According to the results of this in vitro study, it is worth noting that future studies can examine the effect of acidic drinks and thermocycling on the color stability, surface quality and surface topography of different materials after staining with different staining kits.

In this study there are several limitations: it is an in vitro study, the calculation of the light reflection% represent a single value and cannot describe the differences between the different wavelength (300–900 nm) for each measurement, surface roughness was not measured at baseline and the thickness of the staining is manual and not perfectly controlled.

## 5. Conclusions

With the limitations of this in vitro study we suggest that:There is a difference between the materials regarding light reflection%, as the monolithic zirconia showed higher light reflection% when compared to lithium disilicate throughout the entire experiment.For zirconia, both staining kits can be used, but it is important to note that the light reflection% for kit 1 is significantly lower when compared to kit 2.For lithium disilicate, we recommend using kit 1 as we found that kit 2 was partially removed after thermocycling.

## Figures and Tables

**Figure 1 materials-16-02057-f001:**
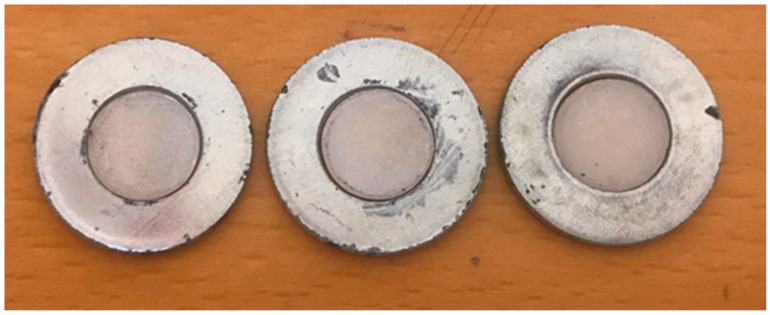
Specimens’ discs placed in a custom-made holder fit for the spectrophotometer machine.

**Figure 2 materials-16-02057-f002:**
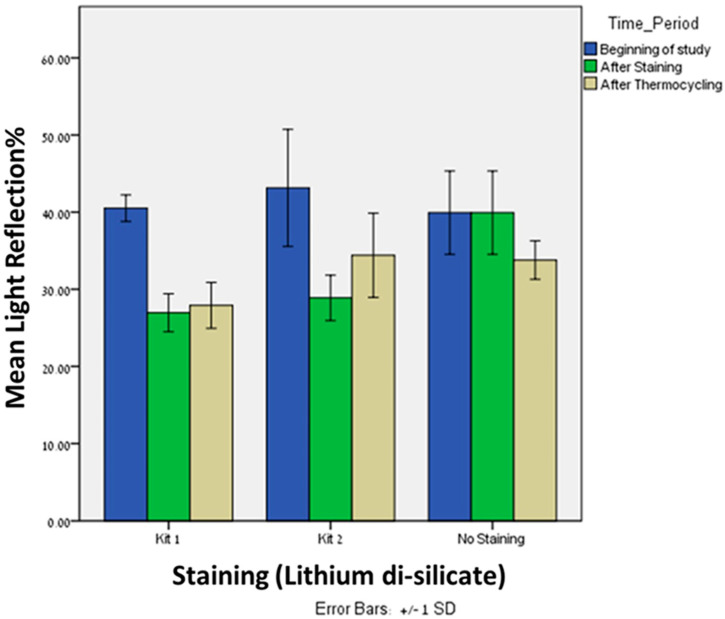
The Mean and SD (±1) of light reflection% of lithium disilicate material after staining with Kit 1 and Kit 2 at three time periods: Beginning of study, after staining and after thermocycling.

**Figure 3 materials-16-02057-f003:**
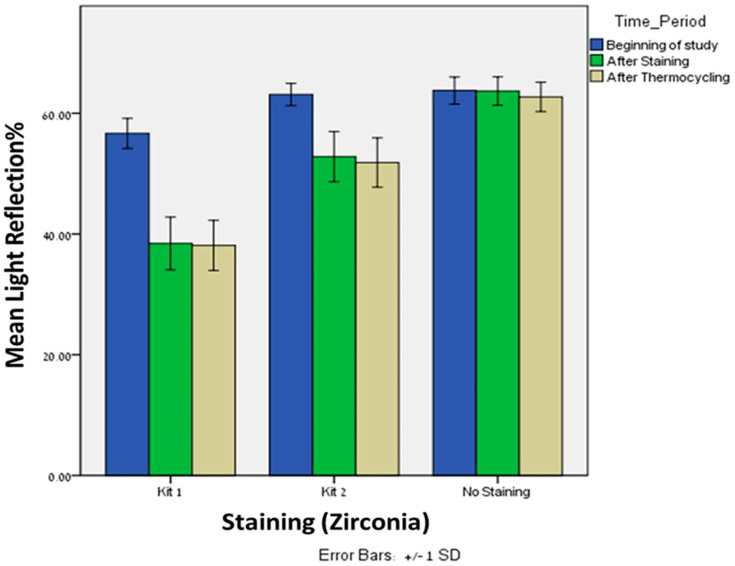
The Mean and SD (±1) of light reflection% of Zirconia material after staining with Kit 1 and Kit 2 at three time periods: Beginning of study, after staining and after thermocycling.

**Table 1 materials-16-02057-t001:** Experimental Groups divided by restoration material and staining kits.

Controlled Group (N = 40)	Kit 1 (N = 40)	Kit 2 (N = 40)
Twenty Monolithic Zirconia discs which remained unstained as baseline	Twenty Monolithic Zirconia discs stained by using the IPS Ivocolor Stain and Glaze-Crystal Stain Kit	Twenty Monolithic Zirconia discs stained by using the Dentsply Sirona Stain and Glaze Kit
Twenty lithium disilicate discs which remained unstained as baseline	Twenty lithium disilicate discs which were stained by using the IPS Ivocolor Stain and Glaze-Crystal Stain Kit	Twenty lithium disilicate discs which were stained by using the Dentsply Sirona Stain and Glaze Kit

**Table 2 materials-16-02057-t002:** Percentile 25 (P25), Median (P50) and percentile 75 (P75) of lithium disilicate and zirconia after staining with kit 1 and kit 2 at three time periods: the beginning of study (T0), after staining (T1) and after thermocycling (T2).

Light Reflection%	Lithium Disilicate			Zirconia		
	No staining	Kit 1	Kit 2	No staining	Kit 1	Kit 2
	(control)			(control)		
P25	38.45 (T0)	39.13 (T0)	39.14 (T0)	62.25 (T0)	54.78 (T0)	62.20 (T0)
	38.45 (T1)	25.01 (T1)	26.77 (T1)	62.25 (T1)	35.75 (T1)	49.67 (T1)
	32.58 (T2)	26.33 (T2)	29.76 (T2)	60.95 (T2)	36.24 (T2)	50.56 (T2)
P50	40.74 (T0)	40.68 (T0)	40.90 (T0)	63.92 (T0)	57.17 (T0)	63.10 (T0)
	40.74 (T1)	26.89 (T1)	28.91 (T1)	63.92 (T1)	39.09 (T1)	53.19 (T1)
	33.63 (T2)	27.17 (T2)	35.60 (T2)	63.05 (T2)	38.62 (T2)	52.90 (T2)
P75	43.54 (T0)	42.02 (T0)	42.05 (T0)	65.79 (T0)	58.49 (T0)	64.61 (T0)
	43.54 (T1)	28.35 (T1)	30.48 (T1)	65.79 (T1)	41.41 (T1)	55.75 (T1)
	35.31 (T2)	30.50 (T2)	39.44 (T2)	64.27 (T2)	40.78 (T2)	54.04 (T2)

## Data Availability

The data presented in this study are available on request from the corresponding author.

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
