# Peer review of "The Light Reflection Changes of Monolithic Zirconia and Lithium Disilicate after Using Two External Staining Kits following by Thermocycling"

_materials, 2023, doi:10.3390/ma16052057_

Round 1

Reviewer 1 Report

The purpose of this in-vitro study was to evaluate the changes of light reflectance% on two materials after using two external staining kits following by thermocycling.

 Please find some limiting point:

1 The sample size has not been justified.

2 The experience of the dental technician must be reported. It's not clear for me how the authors were able to standardize/measure homogeneity and thickness of the staining layers, which may be an important control variable.

3 When the sample size exceeds 50, it may be more appropriate to use K-S test for normality test. If the data is not normally distributed, the use of median and IQR may better reflect the overall situation than the mean and standard deviation.

4 Why did the authors not measure surface roughness? Please provide more explanation and discussion for the results.

5 How does light reflection impact color value and chroma? Can you please also provide small discussion regarding this point.

6 In line 205, n=60 or 120? It’s confusing.

7. What mechanism affects light reflectivity after staining, and are the mechanisms of the two groups different?

8. Please simplify the description of the results.

9. The assumption regarding the coefficient of thermal expansion (CTE) is unsupported. Please provide evidence to support your statement.

Author Response

Thank you very much for the review

Reviewer 2 Report

The article as presented is not of interest to the readers of the journal Materials. The article should be radically revised in accordance with the above comments. 

Comments

1. It is not clear from the text of the article why it is necessary to carry out research related to measurements of light reflectance in zirconium dioxide and lithium disilicate monolith materials and what it gives. Specify in the text of the article.

2. In which devices and industries, it is planned to use these reflective elements. Why is it necessary to carry out this research. What is the problem to be solved. Specify in the text of the article.

3. Why monolithic zirconium dioxide and lithium disilicate were chosen as starting materials for research. Specify in the text of the article.

4. What is the scientific novelty of the research? The scientific substantiation of monolithic zirconium dioxide and lithium disilicate application for reflective elements. Specify in the text of the article.

5. What is the effect of thermal cycling of materials on the properties of monolithic zirconium dioxide and lithium disilicate? Specify in the text of the article.

6. Insert in the text of the article the diffractograms of zirconium dioxide and lithium disilicate materials before and after thermocycling. It is necessary to present the deciphering of all phases included in studied materials. Specify in the text of the article.

7. The authors need to conduct structural studies of the materials, what are the structures of the materials, grain size, etc. Insert the structures of zirconium dioxide and lithium disilicate materials in the text of the article. How thermal cycling affects the structure of materials. Specify in the text of the article.

8. Identify the material properties of monolithic zirconium dioxide and lithium disilicate (specify in the table and draw graphs), based on the area of industry where these materials will work and function. 

9. Compare the properties of the studied zirconium dioxide and lithium disilicate materials with other similar materials already working in industry.

Author Response

Thank you very much for the review

Reviewer 3 Report

Authors attempted to describe Impact of the light reflectance changes of monolithic zirconia and lithium di-silicate after using two external staining kits and after thermocycling. There are few comments based on the current manuscript.

1)     Comment 1: The title of research topic has two ‘AND’ word in it. Please, correct the title. The research topic has been described properly by the authors.

2)     Comment 2: The figure-1 has also been given to another figure. Please, correct the figure numbers in the manuscript.

3)     Comment 3: The construction of the sentences in the manuscript are in proper flow. The readers will appreciate the reading of the manuscript.

4)     Comment 4: The subject area is interesting with all evidence with nicely presentation of tables and figures.

5)   Comment 5: The points in the results section should be highlighted with point number. (as 3.1, 3.2, 3.3)

6)     Comment 6: The conclusion is presented well.

Author Response

Thank you very much for the review

Reviewer 4 Report

Dear Authors, this paper about The light reflectance changes of monolithic zirconia and and

lithium di-silicate after using two external staining kits and after thermocycling is really interesting and i am pretty sure that, once accepted, it will help dental professionals and researchers in their job.

Overall the manuscript is well written and the design of the study is well done.

Some small issues need to be solved before its final publication in the journal.

Abstract: 

  1. Clarify and simplify the language used in the abstract.
  2. Provide a clearer and more concise summary of the results and conclusions at the end of the abstract.

Introduction: You need to improve this part, especially talking more about the consequences caused by grinding and polishing these this materials: here you can find a reference that will help you: Ludovichetti FS, Trindade FZ, Adabo GL, Pezzato L, Fonseca RG. Effect of grinding and polishing on the roughness and fracture resistance of cemented CAD-CAM monolithic materials submitted to mechanical aging. J Prosthet Dent. 2019 May;121(5):866.e1-866.e8.

Materials and methods: well designed and easy to understand

Results: ok

Discussion: discussion is maybe the most important part of an article, and in this case you did not explore this part. You need to improve discussion adding more references and comparing more your results with the already present studies in literature.

Author Response

Thank you very much for the review

Reviewer 5 Report

Dear Authors,

 This study examined the effects of external staining and thermocycling on the light reflectance of monolithic zirconia and lithium disilicate. While the paper provides some useful information, it contains the following concerns.

 Major concerns

The staining methods should be explained in detail because the thickness of the staining layer could affect the light reflectance of both materials. How was the thickness of the staining layer controlled?

The number of thermocycles also might affect the light reflectance of both materials. How did the authors decide the number of thermocycles to 10,000?

Since the Wilcoxon Signed Ranks test is used for comparison between two groups, this reviewer thinks it is inappropriate to use the Wilcoxon Signed Ranks test for statistical analysis of comparisons among T0, T1 and T2. The Friedman Test should be applied for comparisons among the 3 groups (T0, T1 and T2).  

Figures 1, 2 and 3 (light reflectance measurement results) are not easy to understand for this reviewer. This reviewer considers that the authors should create a bar chart for each restoration material, and that each bar chart should display a bar with the mean and SD of the T0, T1 and T2 groups side by side for each staining material.

Figure 1 shows the mean and SD of the light reflectance of 60 specimens for each restoration material before staining and thermocycling. This reviewer recommends eliminating this Figure because the authors divided the 60 specimens into 3 groups of 20 specimens.

In the Introduction section, the authors should emphasize the clinical significance of light reflectance of tooth color restoration materials in more detail.

In the Result section, the explanations for the results of statistical comparison between experimental groups were complicated and easy to understand. The authors should simply describe the results of statistical analysis to obtain easy understand. 

In the Discussion section, the authors should discuss reasons concerning the decrease of light reflectance after staining and the difference on light reflectance between staining kits in more detail.

Minor concerns

The descriptions of “light reflectance” and “light reflection” were mixed in the text. The authors should be consistent with either description.

In the Abstract, the description of the null hypotheses is unnecessary.

Two Figure 1 was found in the text. The first Figure 1 may be unnecessary.

The authors should show the more information about used materials such as main composition, lot (batch) number, manufacturer in Table.

Author Response

Thank you very much for the review

Round 2

Reviewer 2 Report

The authors have submitted a draft with corrections and clarifications in accordance with my comments. But the authors of the article cannot fully (exhaustively) respond to some of the comments. Note on the evaluation of the (mechanical) properties of the studied materials. Apparently not enough experiments, maybe the equipment for conducting experiments, to conduct mechanical tests. I recommend the editor to accept the article as presented.

Reviewer 5 Report

The original manuscript was well revised according to the reviewers' comments. This revision was almost ready to publish.